# Sustainable Immobilization of β-Glucosidase onto Silver Ions and AgNPs-Loaded Acrylic Fabric with Enhanced Stability and Reusability

**DOI:** 10.3390/polym15224361

**Published:** 2023-11-09

**Authors:** Yaaser Q. Almulaiky, J. Alkabli, Reda M. El-Shishtawy

**Affiliations:** 1Department of Chemistry, College of Science and Arts at Khulis, University of Jeddah, Jeddah 21921, Saudi Arabia; 2Department of Chemistry, College of Science and Arts at Alkamil, University of Jeddah, Jeddah 23218, Saudi Arabia; jabdalsamad@uj.edu.sa; 3Chemistry Department, Faculty of Science, King Abdulaziz University, Jeddah 21589, Saudi Arabia; relshishtawy@kau.edu.sa

**Keywords:** β-Glucosidase, immobilization, polyacrylonitrile, silver nanoparticles, reusability

## Abstract

Modified polymer design has attracted significant attention for enzyme immobilization, offering promising applications. In this study, amine-terminated polymers were synthesized by incorporating functional groups into polyacrylonitrile using hexamethylenediamine. This work highlights the successful enzyme immobilization strategy using modified polymers, offering improved stability and expanded operational conditions for potential biotechnological applications. The resulting amino groups were utilized to capture silver ions, which were subsequently converted to silver nanoparticles (AgNPs). The obtained materials, AgNPs@TA-HMDA (acrylic textiles coated silver nanoparticles AgNPs) and Ag(I)@TA-HMDA (acrylic textiles coated with Ag ion) were employed as supports for β-glucosidase enzyme immobilization. The highest immobilization yields (IY%) were achieved with AgNPs@TA-HMDA at 92%, followed by Ag(I)@TA-HMDA at 79.8%, resulting in activity yields (AY%) of 81% and 73%, respectively. Characterization techniques such as FTIR, FE-SEM, EDX, TG/DTG, DSC, and zeta potential were employed to investigate the structural composition, surface morphologies, elemental composition, thermal properties, and surface charge of the support materials. After 15 reuses, the preservation percentages decreased to 76% for AgNPs@TA-HMDA/β-Glu and 65% for Ag(I)@TA-HMDA/β-Glu. Storage stability revealed that the decrease in activity for the immobilized enzymes was smaller than the free enzyme. The optimal pH for the immobilized enzymes was broader (pH 5.5 to 6.5) compared to the free enzyme (pH 5.0), and the optimal temperature for the immobilized enzymes was 60 °C, slightly higher than the free enzyme’s optimal temperature of 50 °C. The kinetic analysis showed a slight increase in Michaelis constant (Km) values for the immobilized enzymes and a decrease in maximum velocity (Vmax), turnover number (Kcat), and specificity constant (Kcat/Km) values compared to the free enzyme. Through extensive characterization, we gained valuable insights into the structural composition and properties of the modified polymer supports. This research significantly contributes to the development of efficient biotechnological processes by advancing the field of enzyme immobilization and offering valuable knowledge for its potential applications.

## 1. Introduction

Biocatalysts have a rich history of use in various scientific and industrial sectors, including bulk and fine chemicals, food, pharmaceuticals, cosmetics, textiles, pulp, and paper [1,2]. In the dairy industry, lactose is a significant byproduct that can be transformed into more valuable and beneficial products such as galactooligosaccharides (GOS) using transgalactosylation processes with retaining β-glucosidases [EC 3.2.1.21] [3]. β-Glucosidases are highly promising in several industrial fields, owing to their exceptional characteristics, such as high efficiency and specificity, which surpass general chemical catalysts. These enzymes can be utilized in various applications such as bioprocessing of biomass, flavor production, the food industry, and the biomedical industry [4,5]. Due to their ability to catalyze reactions under mild conditions, with high selectivity and specificity, β-glucosidases have emerged as an attractive alternative to conventional chemical catalysts. Therefore, the potential for β-glucosidases to enhance the value of inexpensive and abundant raw materials such as lactose, and to generate high-value products, makes them a promising tool for various industrial applications.

To increase the viability of employing enzymes in industrial applications, several efforts have focused on enhancing their stability. Directed evolution, site-directed mutagenesis, enzyme fusion, surface display, and enzyme immobilization are just a few of the approaches utilized to enhance the capabilities of enzymes [6,7]. Among these strategies, enzyme immobilization on solid supports is considered the most successful for enhancing enzyme activity, affinity, selectivity, and stability [8]. Enzyme recycling through immobilization can reduce the cost of producing enzymes and eliminate time-consuming steps such as enzyme preparation [9]. Despite being an interesting subject of several research studies, few immobilization approaches have been effectively marketed, making it a topic of ongoing interest to researchers. Meanwhile, various materials have been considered as supports for enzyme immobilization [10,11,12,13]. The immobilization of catalysts requires intricate chemical processes, including specialized binders and resource-intensive procedures, which can make the resulting catalysts a costly alternative to typical or no-catalyst recovery systems [14]. Typically, the cost of designing and preparing a support matrix is higher than the cost of the actual catalyst in a biocatalyst immobilization procedure. Plus, it is true for a variety of noble and enantioselective organic and inorganic catalysts [15]. In light of these factors, scientists are interested in finding affordable, reliable, and easily accessible support materials for catalyst immobilization. In recent years, there has been significant interest in organic-inorganic hybrid nanomaterials as promising platforms for enzyme immobilization. These nanoparticles have gained attention due to their larger surface-to-volume ratio compared to spherical nanoparticles, as well as their straightforward, environmentally friendly, and cost-effective synthesis methods [16,17]. Several reports have introduced acrylic textiles as an effective support for enzymes [18,19,20]. Acrylic fabric is a desirable support material for enzymes due to its mechanical and chemical stability, large surface area, high enzyme loading capacity, cost-effectiveness, and ease of modification. Acrylic fabric can withstand mechanical and chemical stress, making it a reliable material for enzyme immobilization [21]. Additionally, its porous structure provides a large surface area and easy accessibility to enzyme active sites, allowing for efficient immobilization of enzymes [19]. Additionally, acrylic fabric has a high enzyme loading capacity and is cost-effective as it is readily available. It can also be easily modified with functional groups like amino and carboxyl groups to enhance interaction with enzymes and improve immobilization efficiency. Furthermore, acrylic fabric is compatible with various nanoparticles, including silver nanoparticles, which can be utilized to further enhance enzyme stability and activity [22]. The combination of modified acrylic fabrics with silver nanoparticles and silver ions can lead to an increased effective surface area available for enzyme immobilization. Silver nanoparticles, known for their high surface area-to-volume ratio, can provide additional binding sites for the enzyme molecules. This increased surface area facilitates higher enzyme loading, resulting in improved catalytic efficiency and productivity.

In recent years, inorganic nanoparticles have emerged as promising candidates for enzyme immobilization, offering unique advantages such as increased surface area, enhanced stability, and improved catalytic activity [23]. Among these inorganic nanoparticles, silver nanoparticles (AgNPs) have garnered significant interest. AgNPs possess several properties that make them attractive for enzyme immobilization [24]. AgNPs have been reported to exhibit excellent antimicrobial properties, which can prevent microbial contamination during the immobilization process and subsequent industrial operations [25]. This attribute ensures the longevity and reliability of the immobilized enzyme system. AgNPs have a high surface area-to-volume ratio, enabling higher enzyme loading and facilitating efficient enzyme-substrate interactions. Additionally, AgNPs can stabilize enzymes by providing a protective environment, shielding them from environmental factors such as temperature, pH, and inhibitors [26]. Previous studies have demonstrated the effectiveness of AgNPs for enzyme immobilization, including the immobilization of various enzymes such as lipases, and proteases. These studies have reported enhanced enzyme stability, improved catalytic activity, and increased reusability, highlighting the potential of AgNPs as a valuable tool for enzyme immobilization [27,28].

This study focuses on immobilizing β-glucosidase as a model enzyme on modified acrylic fabrics combined with silver nanoparticles. To achieve this, the outer surface of the acrylic fabric was treated with hexamethylenediamine (HMDA) and then coated with silver nanoparticles (AgNPs) and silver ions (Ag^+^). Subsequently, β-glucosidase was immobilized onto the acrylic. Various analytical techniques, such as FE-SEM, FTIR, DSC, TG/DTG, and Zeta potential, were employed to examine the modified acrylic fabric before and after the enzyme immobilization process. Additionally, the enzyme’s properties, such as pH, temperature, and kinetic behavior, were also studied.

## 2. Methods and Materials

The acrylic fabrics (1/1 woven acrylic (40.6 × 40.6 threads inch-1 for both weft and warp with densities of 0.36 g cm^−3^) were supplied by Misr El-Mahalla Co., Cairo, Egypt. β-Glucosidase from almonds, p-nitrophenyl-β-D-glucopyranoside (pNPG), silver nitrate, hexamethylenediamine (HMDA) (ReagentPlus, 60%), and 1,4-dioxane were acquired from Sigma-Aldrich, Burlington, MA, USA.

## 3. Treatment of the Acrylic

The pristine acrylic fabric underwent a gentle cleaning process using water and acetone, followed by air drying. Two grams of acrylic fabric were mixed with 40 mL of dioxane solvent and agitated, after which 60 mL of HMDA was added. Subsequently, 0.5 g of sodium carbonate was introduced to the reaction mixture, which was then refluxed at 110 °C for 3 h. The treated sample was rinsed with distilled water and acetone before being air-dried. The treated acrylic fabrics, referred to as TA-HMDA, were divided into two groups. In the first group, the fabric was immersed in a 0.05 M silver nitrate solution overnight, rinsed with distilled water, and treated with 5% glucose at 70 °C for one hour. It was then rinsed with distilled water and air-dried. The resulting sample was given the label AgNPs@TA-HMDA. The second half underwent an overnight treatment with 50 mM silver nitrate at 70 °C, followed by a distilled water wash and air drying. The resulting sample was assigned the label Ag(I)@TA-HMDA.

## 4. Enzyme Immobilization

β-Glucosidase enzyme (5 mg, 100 units) was immobilized on both material carriers (AgNPs@TA-HMDA and Ag(I)@TA-HMDA) in various pH media (0.05 M sodium acetate buffer pH 6.0, sodium phosphate buffer pH 7.0, or Tris–HCl buffer pH 8.0). At room temperature, the immobilization procedure was carried out end-over-end for 12 h at 90 rpm. Following a wash with the same buffer, the modified acrylic fabric was left to air dry. The Bradford method was used to quantify the protein concentration, and bovine serum albumin was used as the standard [29]. According to the following equations, the immobilization efficiency, yield, and recovered activity were calculated:Immobilization Yield (IY%) = [Amount of protein Introduced-Protein in the supernatant/Amount of protein introduced] ∗ 100
Activity yield (AY%) = [Immobilized enzyme activity/Initial enzyme activity] ∗ 100

## 5. β-Glucosidase Activity Assay

To determine the activity of β-glucosidase, Narasimha’s method based on the release of p-nitrophenol using the substrate p-NPG was employed [30]. In summary, either 20 mg of immobilized β-glucosidase or 0.1 mL of free β-glucosidase enzyme was added to 0.9 mL of p-NPG (5 mM prepared in 0.1 M sodium acetate buffer with a pH of 6.0). The reaction mixture was then incubated at 37 °C for 5 min, after which 0.5 mL of a 1M Na_2_CO_3_ solution was added to halt the reaction. The absorbance at 405 nm was measured, and the β-glucosidase activity was determined. One unit of β-glucosidase activity was defined as the amount of enzyme required to produce 1 μmoL of p-nitrophenol per minute.

## 6. Characterization of Modified Acrylic Fabrics

The morphology of the samples was examined using a Quanta FEG 250 Field-emission scanning electron microscope (FE-SEM) (FEI Company, Eindhoven, The Netherlands). Energy-dispersive X-ray spectroscopy (EDX) was employed at 20 kV to analyze the metal composition of the treated acrylic fabrics. The functional groups of the samples were investigated using a PerkinElmer Spectrum 100 Fourier-transform infrared spectroscopy (FTIR) instrument (PerkinElmer Life and Analytical Sciences, Shelton, CT, USA). The zeta potential of the samples was estimated using dynamic light scattering with an Entgris Nicomp Nano Z3000 instrument located in Billerica, MA, USA. The thermal properties of the samples were studied using thermogravimetric-derivative and differential scanning calorimetry analysis (TGA-DTG and DSC) conducted with a Shimadzu DTA/TGA-50 instrument from Kyoto, Japan. The heating rate during the analysis was set at 10 °C per minute under a nitrogen atmosphere.

## 7. Optimization of pH and Temperature

The influence of pH and temperature on the activity of β-glucosidase was studied at various pH (0.5 M sodium acetate buffer pH 4.0–6.0; sodium phosphate buffer 6.5–8.0; Tris-HCl buffer 8.5–9.0) and in the range temperature of 30–90 °C. Standard assay conditions were used to determine the optimum pH for free and immobilized β-glucosidase activity, and optimal temperatures were determined under optimal pH and incubated at various temperatures.

## 8. Kinetic Parameters Study

Using different concentrations of p-NPG, the Lineweaver–Burk equation, as shown below, was used to determine the apparent Michaelis–Menten constant ($K_{m}$) and maximum reaction velocity ($V_{max}$) values. The turnover constant (*K_cat_*) and catalytic coefficient ($K_{cat}$/$K_{m}$) were then calculated.
(1)$$\frac{1}{V_{0}}=\frac{K_{m}}{V_{max}}\;\frac{1}{\left[S\right]}+\frac{1}{V_{max}}\;$$
(2)$$V_{max}=K_{cat}\;E_{0}$$
where *V*_0_ is the initial rate of reaction, *E*_0_ is the initial enzyme concentration, and [*S*] is substrate concentration. 

## 9. Reuse and Stable Operation

By monitoring the activity of the immobilized β-glucosidase for 15 recyclers, its reusability was evaluated. The supernatant was decanted following each cycle, and enzyme activity was determined. After being washed with 0.1 M acetate buffer, pH 6.0, immobilized β-glucosidase was put back into the fresh substrate solution. The initial enzyme activity was taken as 100% for better comparability.

By estimating the residual percentage of enzymatic activity of each sample maintained at 4 °C for 8 weeks, the storage stability of free and immobilized β-glucosidase was assessed. The original enzyme activity was set to 100%, then relative activity was contrasted for better comparability.

## 10. Results and Discussion

### 10.1. Material Supporter Fabrication and Enzyme Immobilization

In this study, amine-terminated polymers were synthesized by introducing functional groups to polyacrylonitrile through the addition of hexamethylenediamine (HMDA). These amino functional groups were employed for the capture of silver ions, which were subsequently converted into silver nanoparticles (AgNPs) through oxidative polymerization with glucose. The treatment process of acrylic and the trapping of silver ions and/or AgNPs on the treated acrylic (TA-HMDA) is illustrated in Figure 1. The amine-terminated polymers were able to effectively bind to the silver ions, creating a spacer arm between the polymer and the metal particles. The resulting TA-HMDA material exhibited enhanced properties such as increased surface area and thermal stability, making it an ideal candidate for enzyme immobilization. HMDA, with its longer chain length of six methylene units, can act as a more flexible spacer arm. This flexibility of HMDA can improve enzyme accessibility to the substrate and prevent steric hindrance between the enzyme and the support matrix. Additionally, the longer chain length of HMDA may provide more sites for metal ion binding, which can enhance the stability and activity of the immobilized enzyme [31]. In this study, a method was developed to activate fabric by reductively converting nitrile groups to aminoamidine. The effectiveness of this method was demonstrated by a weight gain of 19% following treatment. Furthermore, when TA-HMDA was incubated in silver nitrate at 70 °C overnight, the weight gain was 9.86%. Silver ion loading was then performed from the resulting aqueous solution, and a redox process using glucose as a reducing agent was used to create AgNPs, resulting in an additional weight gain of 10.05%. Figure 1 schematically illustrates the process of creating AgNPs-loaded, activated acrylic fabric. Table 1 and Table 2 present the various pH buffers that were employed during the immobilization process. It is noteworthy that the immobilized enzymes showed their maximum activities at pH 6.0. The highest IY% was achieved with AgNPs@TA-HMDA at 92%, followed by Ag(I)@TA-HMDA at 79.8%, resulting in activity yields (AY%) of 81% and 73%, respectively. Several types of nanomaterials, including carbon nanotubes [32], graphene oxide [33], and metal oxide nanoparticles [22], have been used for enzyme immobilization on acrylic fabrics. These nanomaterials can provide a stable and well-defined surface for enzyme binding, allowing for improved enzyme activity and durability. In a prior study [22], we utilized employed acrylic fabric loaded with Cu(II) and CuNPs for immobilizing α-amylase and observed that the optimal pH for maximum activity of the immobilized α-amylase was 7.0. The results showed that CuNPs@HMDA-TA had the highest immobilization yield of 81.7%, followed by Cu(II)@HMDA-TA with a yield of 71.7%, the corresponding activity yields were 75% and 61%, respectively. The choice of nanomaterials for enzyme immobilization can have a significant impact on the stability, activity, and selectivity of the immobilized enzymes, and the advantages of using silver nanoparticles (AgNPs) and silver ions (Ag^+^) over copper nanoparticles (CuNPs) and copper ions (Cu(II) can depend on the specific application and conditions. One advantage of using AgNPs and Ag^+^ ions is their excellent biocompatibility and low toxicity, which makes them suitable for use in biomedical and food-related applications [34]. In contrast, CuNPs and Cu(II) ions can be toxic to cells and can cause oxidative stress, which can limit their use in some applications. Another advantage of using AgNPs and Ag^+^ ions is their high stability and resistance to oxidation, which can ensure the long-term stability of the immobilized enzymes. This is particularly important in industrial applications where the immobilized enzymes may be exposed to harsh conditions such as high temperature, pressure, and pH. Furthermore, AgNPs and Ag^+^ ions have been shown to have antimicrobial properties, which can help to prevent contamination in food and biomedical applications. In contrast, CuNPs and Cu(II) ions do not have strong antimicrobial properties and may require higher concentrations to achieve similar effects.

### 10.2. ATR-FTIR Analysis

The (ATR-FTIR) spectra of the pristine acrylic, hexamethylenediamine-treated acrylic (TA-HMDA), and silver-loaded samples in both ions (Ag(I)@TA-HMDA) and nanoparticles (AgNPs@TA-HMDA) and their corresponding immobilized ones (Ag(I)@TA-HMDA/β-Glu and AgNPs@TA-HMDA/β-Glu) are shown in Figure 2. As stated in the previous study [21], the pristine acrylic fabric exhibited characteristic peaks at 1070, 1366, 1452, 1733, 2244, and 2934 cm^−1^, corresponding to vibrations of C–H, C-O-C, C–O, CN, and CH_2_ modes. Following the amination process, these peaks underwent shifts, and new bands appeared at 3345 cm^−1^ and 1655 cm^−1^, attributed to the presence of NH_2_ and/or NH groups in the TA-HMDA sample. Additionally, a shoulder at 1557 cm^−1^ was observed, indicating the presence of C–N and NH bonds. Furthermore, in the TA-HMDA sample, a prominent and wide band around 1450 cm^−1^ was observed, resulting from the overlapping bands of various groups such as CH_2_, C–O, and NH groups.

Upon sliver ion loading, the shape, intensity, and wavenumbers of the latter bands have become different, indicating the success of the formation of Ag(I)@TA-HMDA. Similarly, reducing the sliver-loaded sample to AgNPs@TA-HMDA sample resulted in IR changes due to chemical treatments. Enzyme immobilization onto the silver ion-loaded sample (Ag(I)@TA-HMDA/β-Glu) or into the silver nanoparticles-loaded sample (AgNPs@TA-HMDA/β-Glu) resulted in comparative IR changes before and after immobilization, confirming the success of enzyme immobilization.

### 10.3. Morphological Characterization

The pristine acrylic fabric appears smooth and uniform, with a relatively consistent surface texture across the material (Figure 3a). In contrast, the surface of acrylic fabric treated with HMDA appears to be rougher and more irregular, with an increase in surface roughness observed (Figure 3b). This is due to the chemical modification of the surface of the acrylic fabric with HMDA, which creates new surface features and alters the surface morphology. The further treatment of the acrylic fabric with silver nitrate has resulted in the formation of a silver nano-coated layer, as shown in Figure 3c. The SEM image reveals that the surface of the fabric is covered with a layer of silver nanoparticles, which appear as small, densely packed clusters. This coating has a uniform distribution across the surface of the fabric, and the silver nanoparticles appear to be firmly attached to the fabric surface. Based on the SEM image shown in Figure 3e, it appears that there is no significant change in the surface morphology of the treated acrylic fabric as a result of the formation of a silver ion-coated layer. However, the formation of a silver ion-coated layer on the treated acrylic fabric can lead to chemical changes in the surface chemistry that can be detected by FTIR spectroscopy. These changes can manifest as shifts in the positions of specific peaks in the FTIR spectrum, which correspond to different functional groups and chemical bonds present on the surface of the fabric. Based on the SEM images shown in Figure 3d,f, it appears that the immobilization of enzymes on the treated acrylic fabric has led to the agglomeration or clustering of the silver particles that were previously formed on the fabric surface. This agglomeration of the silver particles is likely due to the binding of the enzyme molecules to the surface of the fabric, which can cause the silver particles to come into closer proximity to each other. 

The successful coating of the treated acrylic fabric with AgNPs or Ag(I) ions was confirmed through the utilization of EDX, as illustrated in Figure 4a,b. EDX is an effective analytical technique used to determine the elemental composition of materials, including the identification and quantification of specific elements. The results presented in Figure 4a,b and the accompanying inset tables provide evidence that the treated acrylic fabric has indeed been coated with both AgNPs and Ag(I) ions. The weight percentages of silver on the fabric surface were found to be 9.83% for AgNPs and 39% for Ag(I) ions. These results confirm that the coating process was successful and show a significant amount of silver that has been deposited onto the treated acrylic fabric. 

### 10.4. The Zeta Potential

The electrostatic properties of support materials play a crucial role in enzyme immobilization. By measuring the zeta potential of the support material, researchers can assess its surface charge and compatibility with the enzyme of interest. Matching the electrostatic properties between the enzyme and the support material allows for strong electrostatic interactions, leading to enhanced enzyme immobilization efficiency, stability, and activity [35]. The zeta potential of treated acrylic fabric was investigated both before and after enzyme immobilization to evaluate any changes in surface charge and potential. The results of the zeta potential measurements can provide insight into the effectiveness of the enzyme immobilization process, as well as the impact of the immobilized enzymes on the surface charge of the treated acrylic fabric [36]. The zeta potential values of the different samples under study were measured and presented in Figure 5a,b and the inset data. Figure 5a illustrates the zeta potential measurements of the pristine acrylic fabric, HMDA-treated acrylic fabric, AgNPs-coated treated acrylic fabric, and the immobilized β-Glu on the AgNP-coated treated acrylic fabric. The zeta potential of the pristine acrylic fabric was recorded as −14.69 mV. Following treatment with HMDA, the zeta potential increased to −1.74 mV, indicating the presence of amino groups introduced by the HMDA treatment. Coating the treated acrylic fabric with AgNPs resulted in a decrease in the zeta potential to −5.62 mV, suggesting that the AgNP coating was more attractive to the treated fabric compared to the HMDA treatment alone. Upon immobilization of β-Glu on the AgNP-coated treated acrylic fabric, the zeta potential further decreased to −11.70 mV. This reduction in zeta potential can be attributed to the electrostatic interaction between the ionizable groups of the enzyme (which typically have more anionic groups) and the support material (which typically has more cationic groups) [22]. Such interactions can lead to a decrease in the zeta potential value and can impact the efficiency of the enzyme immobilization process. In Figure 5b, the zeta potential of the Ag(I)@TA-HMDA sample was observed to be −7.53 mV. This value can be attributed to the presence of -OH groups on the surface of the acrylic fabric resulting from hydration [35]. After β-Glu immobilization, the zeta potential value further decreased to −15.69 mV. This observation has suggested that the AgNPs@ TA-HMDA coating is more efficient in binding and immobilizing β-glucosidase, likely due to the presence of the AgNPs, which can provide additional sites for enzyme binding and enhance the electrostatic interactions between the enzyme and the fabric surface [37].

### 10.5. The Thermal Properties

The TG-DTG curves and DSC curves of the system under investigation (as shown in Figure 6, Figure 7 and Figure 8) reveal the thermal decomposition behavior of the samples. Based on the TG curves and confirmed by DTG investigations, it is evident that there are structural transformations occurring during the thermal decomposition process. The decomposition process involves multiple steps and can be divided into distinct stages. The initial stage involves the removal of adsorbed water from the samples, followed by the release of coordination water. These water removal steps are typically observed as weight loss in the TG curves. The dehydration of adsorbed water occurs first, and then coordination water is released from the system. After the water removal stages, the thermal degradation of the organic backbone of the polymer takes place. This stage was characterized by the release of organic fragments as the temperature increased. The organic backbone fragments contribute to the weight loss observed in the TG curves. Upon a thorough analysis of the TG-DTG curves of all the samples (refer to Figure 6 and Figure 7), it is evident that the thermal degradation profile occurs in two phases. The first stage involves the removal of water, both adsorbed and coordinated, while the second stage corresponds to the decomposition of the organic backbone and the release of organic fragments. Table 3 provides a summary of the thermal decomposition data, which likely includes information such as onset temperature, peak temperature, and weight loss at different stages of decomposition for each sample. Based on the information provided in Figure 6a,b and Table 3, the onset degradation temperatures (T_onset_) for different samples were observed. For authentic acrylic fabric, TA-HMDA (treated acrylic fabric with HMDA), AgNPs@TA-HMDA (treated acrylic fabric with HMDA and silver nanoparticles), AgNPs@TA-HMDA/β-Glu, Ag(I)@TA-HMDA, and Ag(I)@TA-HMDA/β-Glu, the T_onset_ values were determined to be 314 °C, 289 °C, 284 °C, 303 °C, 283 °C, and 277 °C, respectively. Furthermore, Table 3 summarizes the temperature at which 50% mass loss (T_50_) was observed for each of the mentioned samples. The data in Table 3 indicates that the presence of AgNPs and Ag(I) ions improved the thermal properties of the treated acrylic fabric. The DTG curves depicted in Figure 7a clearly demonstrate decomposition process steps for all the samples. For authentic acrylic fabric, the decomposition steps were observed within the temperature ranges of 292–370 °C and 385–481 °C, accompanied by two endothermic TD-TG peaks observed at 331 °C and 433 °C. Following chemical treatment, the sample exhibited two distinct decomposition steps within the temperature ranges of 276–345 °C and 387–498 °C, along with endothermic TD-TG peaks observed at 312 °C and 441 °C. In the case of the acrylic fabric coated with AgNPs (AgNPs@TA-HMDA), a single decomposition step occurred within the temperature range of 264–444 °C, accompanied by an endothermic TD-TG peak at 324 °C. After immobilization, the sample displayed a decomposition step within the range of 308–437 °C, with an endothermic TDTG peak observed at 344 °C.

Moving on to Figure 7b, the DTG curve of the acrylic fabric coated with Ag(I) ions revealed two decomposition steps in the temperature ranges of 180–270 °C and 270–387 °C, with endothermic TD-TG peaks observed at 231 °C and 330 °C. After immobilization, the sample showed a decomposition step within the range of 278–506 °C, with an endothermic TD-TG peak at 329 °C. The observed results from the TG-DTG and DSC curves are in accordance with each other, demonstrating consistency in the thermal behavior of the samples. DSC has been widely utilized to study the thermal properties and structures of inorganic-organic compounds, as well as coordination polymers involving transition metal ions [38,39,40]. In this study, DSC was used to investigate the caloric requirements of the support-enzyme system hydrolysis. The thermograms of each sample displayed two distinct endothermic peak regions. The initial combustion region primarily involved the degradation of low-volatile chemicals, while the second region involved the decomposition of more stable volatiles, such as fixed carbon [41]. Figure 8a,b illustrates the DSC curves obtained for all samples, while Table 2 provides information on the onset and end temperatures of the latent heat of fusion, as well as the enthalpy of fusion (∆H). The TG-DTG and DSC curves demonstrate overall agreement in their results. The thermograms of treated samples display broad endothermal bands. These bands can be attributed to the denaturation of random chains and α-helices of the protein at elevated temperatures [42]. The observed enhanced thermal stability in the AgNPs@TA-HMDA/β-Glu and Ag(I)@TA-HMDA/β-Glu samples can be attributed to the strong electrostatic interaction between the enzymes and the functionalized acrylic material. The improved thermal stability of the acrylic network support can be attributed to the increased affinity between the amino groups of the enzymes and the AgNPs or Ag(I) ions. This enhanced affinity leads to a stronger interaction, thereby contributing to the improved thermal stability observed in the system.

### 10.6. Reusability and Storage Stability

The preservation percentages of the AgNPs@TA-HMDA/β-Glu and Ag(I)@TA-HMDA/β-Glu enzymes after multiple reuses provide insights into their stability and effectiveness over time. After ten reuses, the AgNPs@TA-HMDA/β-Glu enzyme retained 87% of its initial activity, while the Ag(I)@TA-HMDA/β-Glu enzyme retained 84% of its initial activity (Figure 9a). These results indicate that both enzymes maintained a relatively high level of activity after repeated use. This suggests that the AgNPs@TA-HMDA and Ag(I)@TA-HMDA coatings provided effective support and stabilization for the β-glucosidase enzyme. The preservation of enzyme activity is crucial for practical applications, as it ensures the enzyme’s longevity and reusability [43]. After 15 reuses, the preservation percentages decreased to 76% for the AgNPs@TA-HMDA/β-Glu enzyme and 65% for the Ag(I)@TA-HMDA/β-Glu enzyme. These results indicate a gradual decrease in enzyme activity over repeated uses. The decline in preservation percentage suggests that the stability and effectiveness of the enzymes may have been compromised to some extent due to factors such as enzyme denaturation, structural changes, or loss of active sites [44,45]. However, it is important to note that the enzymes still retained a significant portion of their initial activity after 15 reuses, indicating their robustness and suitability for repeated applications. The comparison between the AgNPs@TA-HMDA/β-Glu and Ag(I)@TA-HMDA/β-Glu enzymes reveals that the former exhibited better preservation of activity after both ten and 15 reuses. This suggests that the AgNPs@TA-HMDA coating may provide superior stabilization and protection for the β-glucosidase enzyme compared to the Ag(I)@TA-HMDA coating. In another study, it was reported that β-glucosidase immobilized on hydroxyapatite retained 70% of its initial activity after 10 cycles of reuse [46]. Similarly, when β-glucosidase was immobilized on chitosan-MWCNTs, it retained approximately 72.8% of its initial activity after 10 reuses [47]. These findings highlight the potential of these immobilization strategies for preserving enzyme activity over multiple cycles. 

Immobilizing enzymes aim to enhance their stability during storage, as the activity of free enzymes tends to decrease over time. When comparing the activity of free β-glucosidase with AgNPs@TA-HMDA/β-Glu and Ag(I)@TA-HMDA/β-Glu throughout an 8-week storage period at 4 °C, it was observed that the decrease in activity for the immobilized enzymes was smaller than that of the free enzyme. Specifically, the activity of AgNPs@TA-HMDA/β-Glu decreased by 18%, while the activity of Ag(I)@TA-HMDA/β-Glu decreased by 22% (as shown in Figure 9b). According to a previous study, the rate of action of immobilized β-glucosidase on modified polyester fabric decreased by 26% after a storage period of 6 weeks [33]. Using AgNPs@TA-HMDA and Ag(I)@TA-HMDA matrices as supporters for the enzyme can provide a protective environment for the enzyme, shielding it from external factors that may cause denaturation or degradation. This enhanced stability helps to preserve the enzyme’s activity over time.

### 10.7. Effect of pH and Temperature

The optimal pH was determined by conducting experiments with free β-Glu, AgNPs@TA-HMDA/β-Glu, and Ag(I)@TA-HMDA at different pH levels, as shown in Figure 10a. The free β-Glu enzyme showed its highest activity at pH 5.0, while the immobilized enzyme exhibited a broader pH range of 5.5 to 6.5, with the optimal pH being 6.5. However, deviations from the optimal pH can occur due to the strong interaction between the carrier matrix and the enzyme, involving hydrogen bonds and electrostatic reactions. At pH 9, the activity of the immobilized β-Glu on AgNPs@TA-HMDA and Ag(I)@TA-HMDA increased significantly by 4-fold and 3.3-fold, respectively, compared to the free form. The higher pH stability of the immobilized enzyme complexes can be attributed to secondary interactions or diffusion limitations between enzyme molecules and the supporting materials. This finding is consistent with previous research on immobilized β-Glu, where it was immobilized on a combination of magnetic iron oxide nanoparticles coated with chitosan and sodium alginate [3,48].

The effect of temperature on the free and immobilized enzymes was analyzed at temperatures varying from 30 to 80 °C (Figure 10b). The optimal temperature for the free enzyme was found to be 50 °C, while both immobilized enzymes exhibited their highest activities at 60 °C. The free enzyme retained 51% of its initial activity when subjected to a temperature of 80 °C, whereas the immobilized β-Glu on AgNPs@TA-HMDA and Ag(I)@TA-HMDA retained 74% and 68% of their initial activities, respectively. This indicates that the immobilized enzymes showed a higher level of thermal stability compared to the free enzyme. This result is consistent with previous findings reported by Almulaiky [33], Gupta et al. [49], Patel et al. [50], and Verma et al. [51], who also observed that 60 °C was the optimal temperature for immobilized β-glucosidase enzyme.

### 10.8. Kinetic Behavior

As shown in Table 4, the slight increase in Km values observed for AgNPs@TA-HMDA/β-Glu (3.1 mM) and Ag(I)@TA-HMDA/β-Glu (3.7 mM) compared to free β-Glu (2.6 mM) can be attributed to the immobilization process and the interaction between the enzyme and the immobilization matrix. Moreover, this interaction can introduce structural changes or steric hindrance that may affect the accessibility of substrates to the active site of the enzyme [52,53]. Consequently, there may be a slight alteration in the Km value, which was a measure of the substrate concentration at which the enzyme operates at half of its maximum velocity. The change in Km suggested that the immobilization process may have slightly affected the enzyme’s affinity for its substrate, resulting in a higher substrate concentration required to achieve the same catalytic efficiency as the free enzyme. However, it is important to note that the observed differences in Km values are relatively small, indicating that the immobilized enzymes still retain their catalytic activity and can effectively bind and convert the substrate despite the slight alteration in substrate affinity. Alnadari et al. [3] also observed similar results, where the Km value of free β-Glu using pNPG was 2.68 mM, and it increased to 2.84 mM after immobilizing β-Glu on SA-Fe_3_O_4_ MNPs. The changes observed in the Vmax, Kcat, and Kcat/Km values for AgNPs@TA-HMDA/β-Glu and Ag(I)@TA-HMDA/β-Glu compared to free β-Glu can be attributed to the specific characteristics of the immobilization. Immobilization of an enzyme can lead to changes in its catalytic properties due to factors such as altered microenvironment, changes in enzyme conformation, or limitations in substrate accessibility. These factors can affect the enzyme’s turnover rate (Kcat), substrate affinity (Km), and the maximum velocity of the reaction (Vmax) [54]. In the case of AgNPs@TA-HMDA/β-Glu and Ag(I)@TA-HMDA/β-Glu, the decrease in Vmax values (12.7 U/min for AgNPs@TA-HMDA/β-Glu and 9.8 U/min for Ag(I)@TA-HMDA/β-Glu) compared to free β-Glu (17.5 U/min) suggested a reduction in the maximum reaction rate achievable by the immobilized enzymes. This reduction may be due to the physical or structural constraints imposed by the immobilization matrices, which can hinder the movement or accessibility of the substrate to the active site of the enzyme [55]. The decrease in Kcat values (0.818 s^−1^ for AgNPs@TA-HMDA/β-Glu and 0.804 s^−1^ for Ag(I)@TA-HMDA/β-Glu) compared to free β-Glu (0.831 s^−1^) also indicates a reduction in the turnover rate of the immobilized enzymes. Although, this reduction can be attributed to changes in the enzyme’s microenvironment or conformation upon immobilization, which may affect the catalytic efficiency of the active site [56]. The decrease in specificity constant (Kcat/Km) values (264 M^−1^.s^−1^ for AgNPs@TA-HMDA/β-Glu and 217 M^−1^.s^−1^ for Ag(I)@TA-HMDA/β-Glu) compared to free β-Glu (316 M^−1^.s^−1^) suggests a decrease in the catalytic efficiency or substrate binding affinity of the immobilized enzymes. This decrease can be attributed to the alterations in the enzyme’s microenvironment or structural changes induced by the immobilization process [57].

## 11. Conclusions

In conclusion, this study demonstrates the successful use of modified polymers, specifically AgNPs@TA-HMDA and Ag(I)@TA-HMDA, as effective supports for β-glucosidase enzyme immobilization. The immobilized enzymes showed high yields of immobilization and retained significant enzymatic activity. The modified polymers provided improved storage stability, a broader pH range for optimal activity, and elevated optimal temperature compared to the free enzyme. These findings indicate the potential of these modified polymers for various industrial applications requiring enhanced enzyme stability and expanded operational conditions. The study highlights the importance of tailored polymer design and the incorporation of metal nanoparticles for optimizing enzyme immobilization. These findings offer valuable insights for the development of improved enzyme immobilization strategies with enhanced performance and wider applicability. Table 5 shows the advantages and disadvantages of free and immobilized enzymes on fabricated material support.

## Figures and Tables

**Figure 1 polymers-15-04361-f001:**
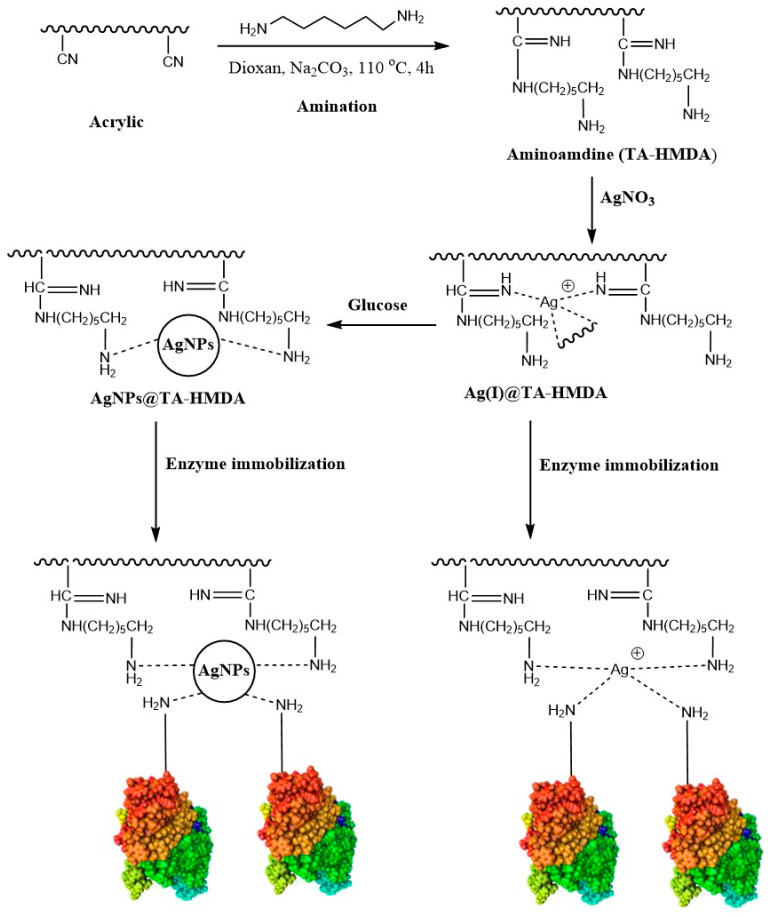
A schematic reorientation for the fabrication of silver ions and AgNPs-loaded acrylic fabric.

**Figure 2 polymers-15-04361-f002:**
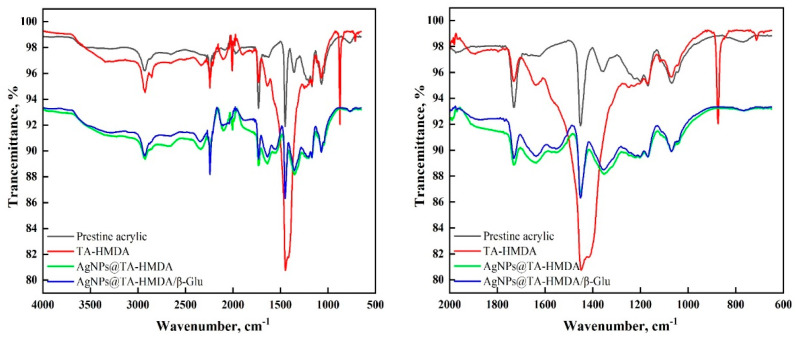
ATR-FTIR spectra of pristine acrylic and treated acrylic before and after enzyme immobilization.

**Figure 3 polymers-15-04361-f003:**
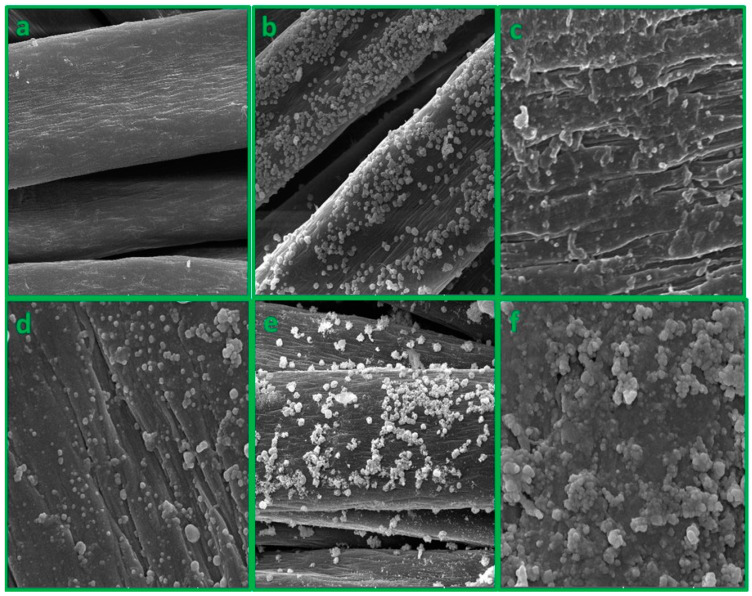
FTSEM of (**a**) Pristine acrylic, (**b**) TA-HMDA, (**c**) AgNPs@TA-HMDA, (**d**) AgNPs@TA-HMDA/β-Glu, (**e**) Ag(I)@TA-HMDA, and (**f**) Ag(I)@TA-HMDA/β-Glu.

**Figure 4 polymers-15-04361-f004:**
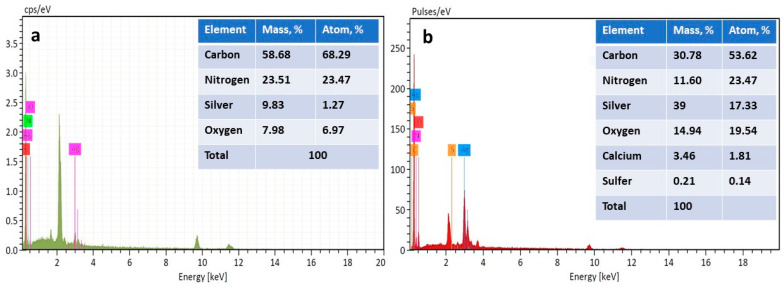
The SEM–energy-dispersive X-ray (EDX) spectra of (**a**) AgNPs@TA-HMDA and (**b**) Ag(I)@TA-HMDA.

**Figure 5 polymers-15-04361-f005:**
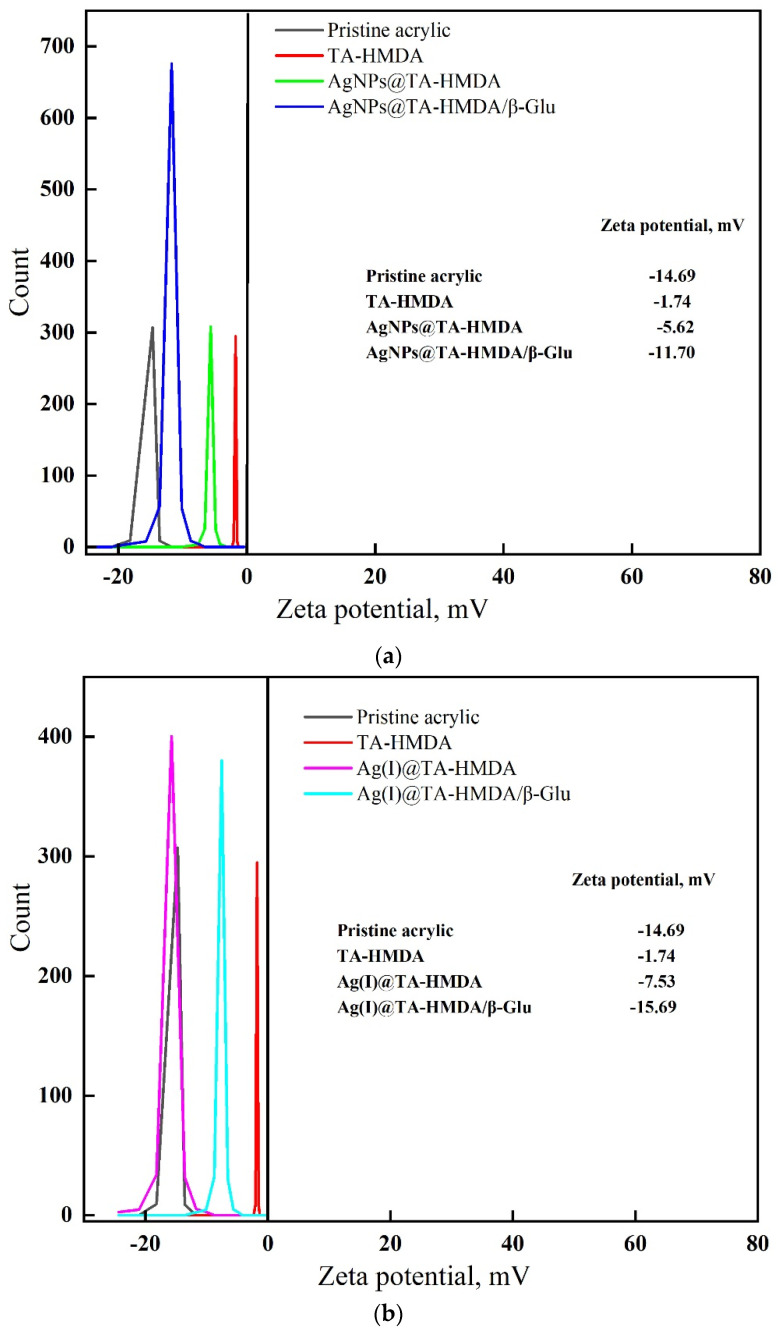
Zeta potential value of pristine acrylic and treated acrylic before and after enzyme immobilization.

**Figure 6 polymers-15-04361-f006:**
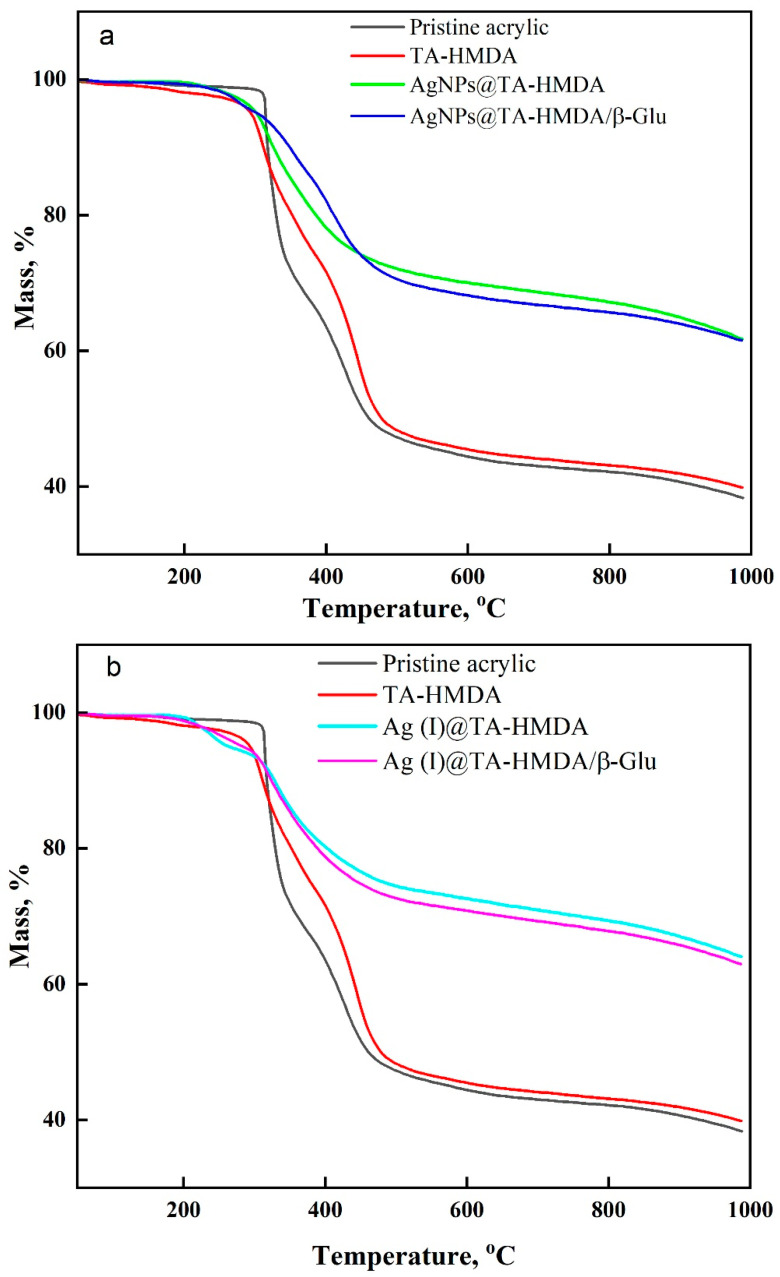
(**a**,**b**) TGA curve of acrylic fabric, modified acrylic fabric, and modified acrylic fabric after modification.

**Figure 7 polymers-15-04361-f007:**
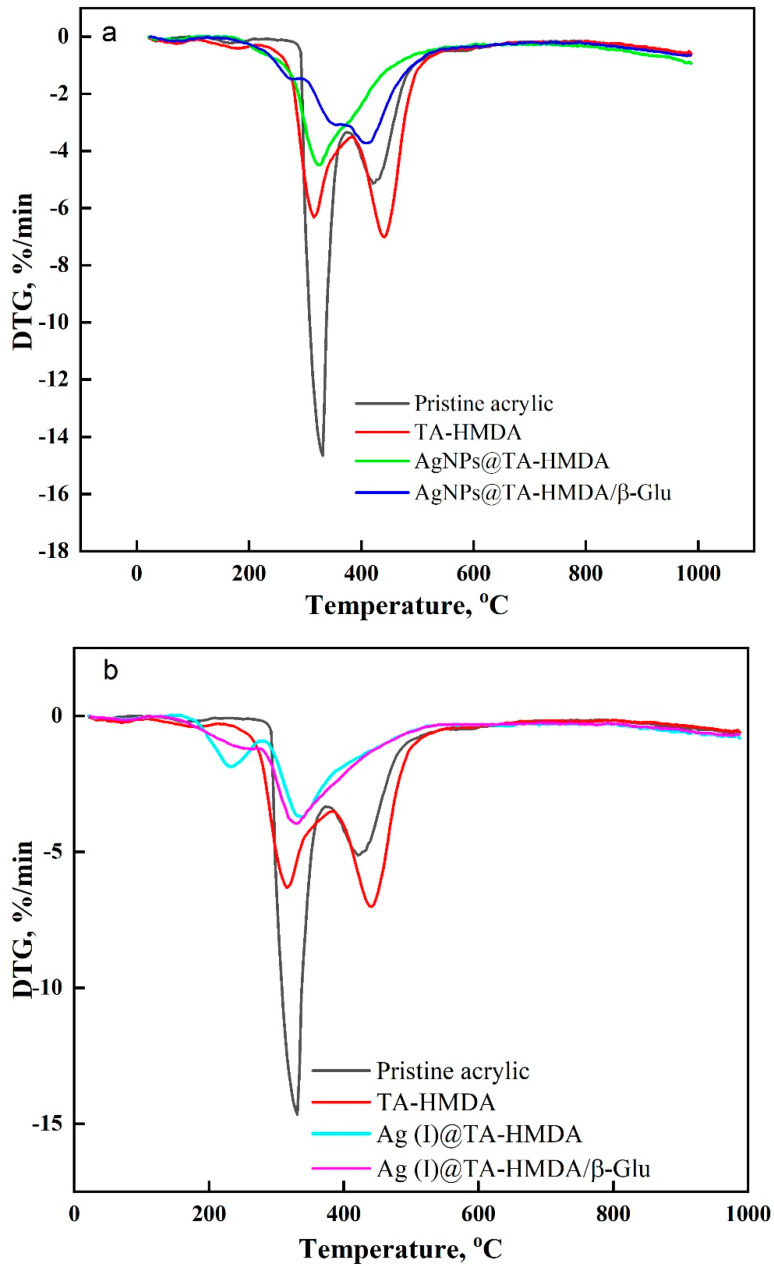
(**a**,**b**) DTG curve of acrylic fabric, modified acrylic fabric, and modified acrylic fabric after modification.

**Figure 8 polymers-15-04361-f008:**
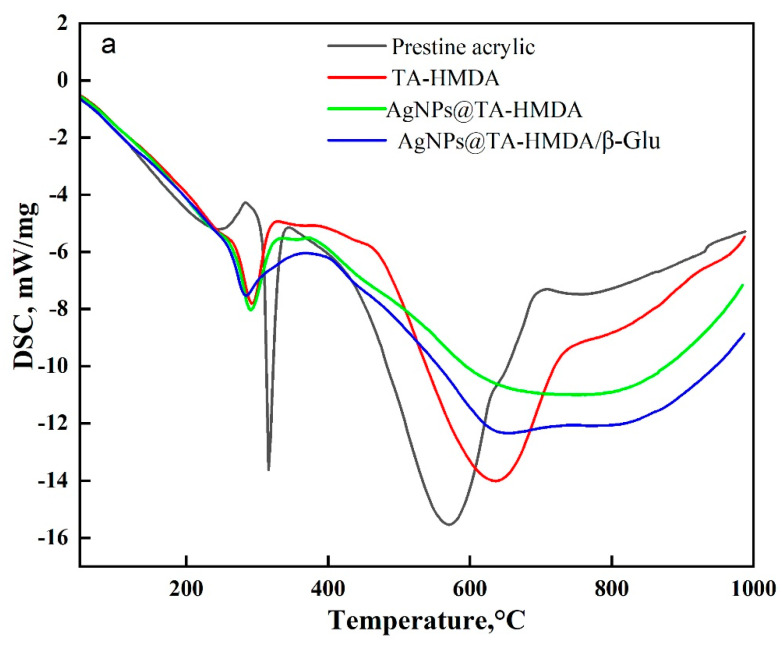
(**a**,**b**) DSC curve of acrylic fabric, modified acrylic fabric, and modified acrylic fabric after modification.

**Figure 9 polymers-15-04361-f009:**
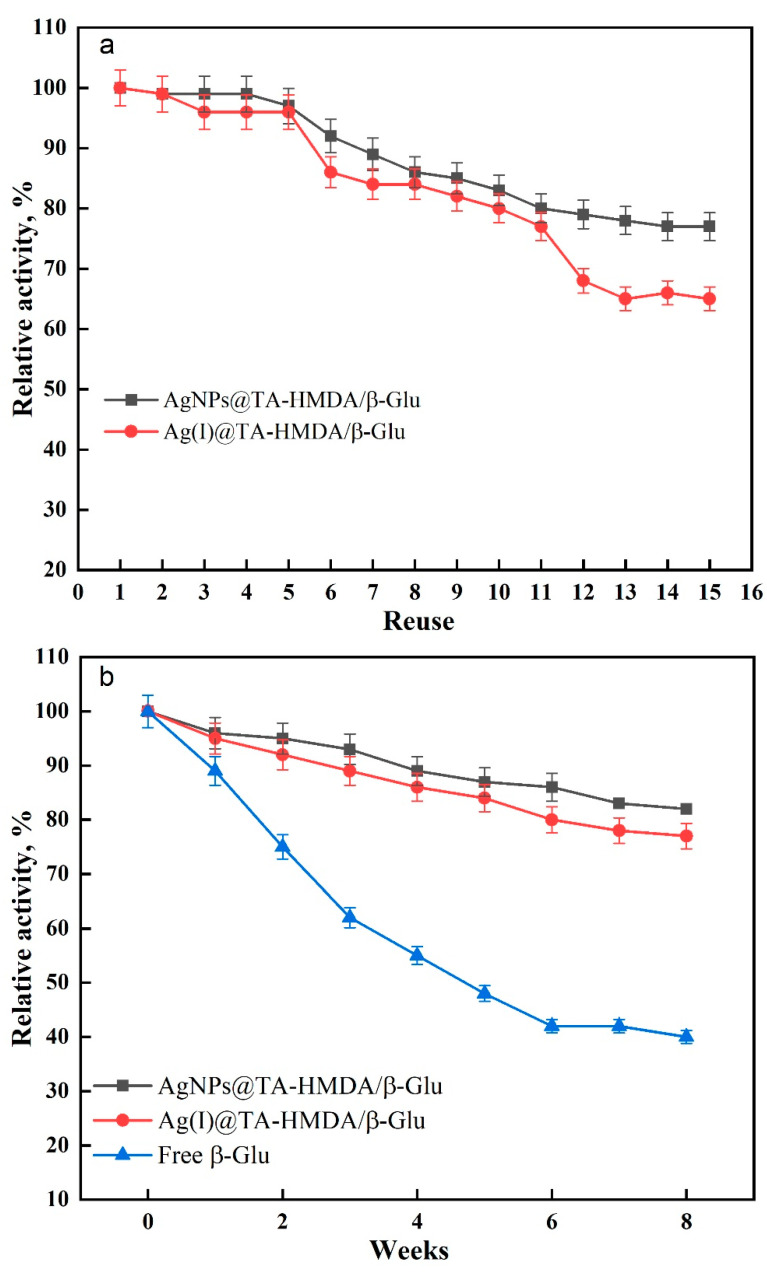
(**a**) reusability, and (**b**) storage stability of β-glucosidase.

**Figure 10 polymers-15-04361-f010:**
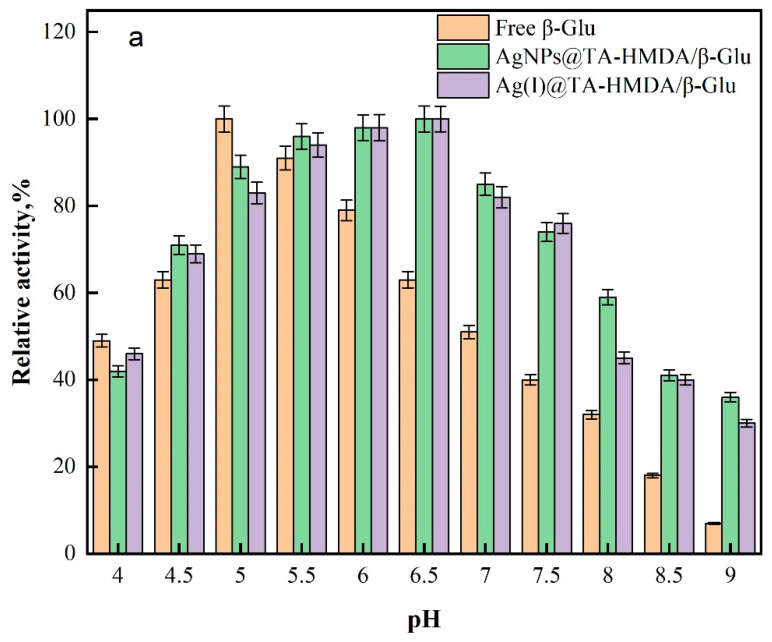
Effect of (**a**) pH and (**b**) Temperature on the activity of β-glucosidase.

**Table 1 polymers-15-04361-t001:** Immobilization yields (IY%) and activity yields (AY of β-glucosidase on AgNPs@TA-HMDA.

AgNPs@TA-HMDA/β-Glu						
pH	Protein Introduced (mg)	Immobilized Protein mg/g Support	IY (%)	Initial Activity (Units)	Immobilized Enzyme Activity (Units)	AY (%)
6		4.6	92		16.2	81
7	5	3.9	78	20	12.4	62
8		2.7	54		9.8	49

**Table 2 polymers-15-04361-t002:** Immobilization yields (IY%) and activity yields (AY of β-glucosidase on Ag (I)@TA-HMDA.

Ag (I)@TA-HMDA/β-Glu						
pH	Protein Introduced (mg)	Immobilized Protein mg/g Support	IY (%)	Initial Activity (Units)	Immobilized Enzyme Activity (Units)	AY (%)
6		3.99	79.8		14.6	73
7	5	3.05	61	20	11	55
8		2.15	43		8.2	41

**Table 3 polymers-15-04361-t003:** Thermogravimetric degradation behavior of pristine acrylic and treated acrylic.

	DSC				TGA-DTG				
	Degradation Process								
Sample	T_Onset_ °C	T_Peak_ °C	T_End_ °C	∆H J/g	T_Onset_ °C	T_50_ °C	T_End_ °C	Inflection Point, °C	Mass Change %
Pristine acrylic	283	316	340	−170	314	343	424	315	50.43
	410	568	688	−1157	-	-	-	-	-
TA-HMDA	261	294	324	−74	289	395	433	309	48.45
	462	624	735	−914	-	-	-	-	-
AgNPs@TA-HMDA	249	291	330	−86	284	359	440	379	26.35
AgNPs@TA-HMDA/β-Glu	251	284	359	−84.8	303	386	473	405	28.29
Ag(I)@TA-HMDA	117	246	273	−8	283	353	392	340	19.13
	273	303	339	−23.9	-	-	-	-	-
Ag(I)@TA-HMDA/β-Glu	143	290	332	−170	277	36 5	400	316	21.75

**Table 4 polymers-15-04361-t004:** Kinetic parameters of free and immobilized enzymes.

	Free Glu	AgNPs@TA-HMDA/β-Glu	Ag(I)@TA-HMDA/β-Glu
Km (mM)	2.6	3.1	3.7
Vmax (U/min)	17.5	12.7	9.8
Kcat (s^−1^)	0.831	0.818	0.804
Kcat/Km (M^−1^.s^−1^)	316	264	217

**Table 5 polymers-15-04361-t005:** The advantages and disadvantages of free and immobilized enzyme on fabricated material support.

Parameters	Free β-Glu	AgNPs@TA-HMDA/β-Glu	Ag(I)@TA-HMDA/β-Glu
Cost	Low	Moderate	Moderate
Viability	Enzyme is free	Enzyme immobilized on support	Enzyme immobilized on support
Stability	Moderate	High	High
Activity	Exhibit loss over time	Enhanced activity	Enhanced activity
Reusability	Limited reusability	Higher reusability	Higher reusability

## Data Availability

The data presented in this study are available on request from the corresponding author.

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
