# Peer review of "Sustainable Immobilization of β-Glucosidase onto Silver Ions and AgNPs-Loaded Acrylic Fabric with Enhanced Stability and Reusability"

_polymers, 2023, doi:10.3390/polym15224361_

Round 1

Reviewer 1 Report

Comments and Suggestions for Authors

Please see the attachement pdf file.

Comments on the Quality of English Language

I suggest to the authors to do some minor editing of English language.

Reviewer 2 Report

Comments and Suggestions for Authors

1. correct "β-. Glucosidase activity assay" in the line no. 119 

2. The authors can add 1-2 lines about significance of measuring zeta potential

3. In many places percentage change in the enzyme activity have been mentioned. It will be good if the authors could mention the activity either in IU or FPU.

4. The authors can include relevant formula or equation to determine the enzyme kinetic parameters.

5. The authors can also include the pros and cons of Free Glu, AgNPs@TA-HMDA/βGlu, Ag(I)@TA-HMDA/β-Glu.  The paper will be very attractive if you could include the cost, viability, stability, activity etc. of Free Glu, AgNPs@TA-HMDA/βGlu, Ag(I)@TA-HMDA/β-Glu in a table.

Round 2

Reviewer 1 Report

Comments and Suggestions for Authors

Dear Authors,

The manuscript has been improved a lot; however it will be suitable for publication in "Polymers" after addressing a missing comment in my first review, which was regaring citing and referring to some recent research articles in general on hybrid systems involving nanomaterials. It is important to mention the importance of hybrid materials in this manuscript. Once the authors addressed this comment, I would be happy to recommend for publication. 

Thanks for your attention

Author Response

Dear Editor and reviewers:

We would like to express our sincere thanks to the editor and reviewers for the constructive and positive comments concerning our manuscript entitled “Enhanced stability and reusability of β-glucosidase immobilized on silver ion and AgNPs-loaded acrylic fabric. Those comments were all valuable and very helpful for revising and improving our paper, as well as the important guiding significance to our research. We have revised our paper in accordance to your comments. In what follows, we would like to response the comments of you and give a detailed account of the changes made to the original manuscript. The comments are reproduced, and our responses are given directly afterward in a different color.

file “Revision Manuscript”.

The manuscript has been improved a lot; however, it will be suitable for publication in "Polymers" after addressing a missing comment in my first review, which was regaring citing and referring to some recent research articles in general on hybrid systems involving nanomaterials. It is important to mention the importance of hybrid materials in this manuscript. Once the authors addressed this comment, I would be happy to recommend for publication. 

Response: we would like to thank the reviewer for this comment. A paragraph about hybrid nanomaterials was mentioned in revised manuscript, in introduction section; lines 71-75.

In recent years, there has been significant interest in organic-inorganic hybrid nanomaterials as promising platforms for enzyme immobilization. These nanoparticles have gained attention due to their larger surface-to-volume ratio compared to spherical nanoparticles, as well as their straightforward, environmentally friendly, and cost-effective synthesis methods [16, 17].
